# A Narrative Review of Biomarkers and Imaging in the Diagnosis of Acute Aortic Syndrome

**DOI:** 10.3390/diagnostics15020183

**Published:** 2025-01-14

**Authors:** Ümit Arslan, Izatullah Jalalzai

**Affiliations:** Department of Cardiovascular Surgery, Faculty of Medicine, Atatürk University, Erzurum 25030, Türkiye; ijalalzai@gmail.com

**Keywords:** acute aortic syndrome, aortic dissection, biomarkers, D-dimer, computed tomography, artificial intelligence

## Abstract

Acute aortic syndrome (AAS) encompasses a range of life-threatening conditions, including classical dissection, intramural hematoma, and penetrating aortic ulcer. Each of these conditions presents distinct clinical characteristics and carries the potential to progress to rupture. Because AAS can be asymptomatic or present with diverse symptoms, its diagnosis requires clinical evaluation, risk scoring, and biomarkers such as D-dimer (DD), C-reactive protein (CRP), homocysteine, natriuretic peptides (BNP), and imaging modalities like computed tomography (CT), magnetic resonance imaging (MRI), and echocardiography. While this review primarily focuses on widely used and clinically accessible biomarkers and imaging techniques, it also discusses alternative biomarkers proposed for diagnostic use. Although CT remains the gold standard for diagnosis, biomarkers facilitate rapid risk stratification, complementing imaging techniques. Emerging technologies, such as metabolomics, are reshaping diagnostic algorithms. Despite advances in diagnostic methods, challenges such as misdiagnosis and missed diagnoses persist. Ongoing research into novel biomarkers and innovative imaging techniques holds promise for improving diagnostic accuracy and patient outcomes.

## 1. Introduction

Aortic dissections and aneurysms have been a focus of scientific and clinical interest for over 250 years, beginning with Dr. Frank Nicholls’ seminal autopsy report on King George II [1]. As the aorta is increasingly recognized as a functional “organ”, advancements in imaging technologies have greatly enhanced the understanding of aortic pathologies. This progress led to the coining of the term acute aortic syndrome (AAS) in the early 2000s [2]. AAS is characterized by distinct clinical entities, including classical dissection (CD), intramural hematoma (IMH), penetrating aortic ulcer (PAU), and incomplete dissection (ID), all of which share a common origin in the disruption of the aortic media. Although these conditions differ pathologically, they share overlapping clinical profiles that can lead to life-threatening complications, such as malperfusion, rupture, and aneurysm formation, necessitating prompt diagnosis and urgent intervention [3,4].

Traumatic aortic injury is included in AAS due to its acute and emergent nature. However, AAS represents only one component of the broader spectrum of “aortic diseases”, which encompasses aortic aneurysms, pseudoaneurysms, atherosclerotic and inflammatory conditions, and genetic and congenital disorders [5]. Originating from the left ventricle and supplying blood to the entire body, the aorta is aptly referred to as the “tree of life”, underscoring its critical role in systemic circulation. Recognizing and addressing aortic pathologies are therefore of paramount importance. This review aims to evaluate the efficacy of laboratory and imaging modalities in the diagnosis of AAS, with a focus on aortic diseases associated with genetic predispositions and aortitis that may lead to AAS.

## 2. Methodology

A narrative literature search was performed using PubMed, Embase, Scopus, and Web of Science to identify studies on AAS, aortic dissection, diagnostic methods, biomarkers, and imaging techniques published between 2000 and 2024. The search focused on studies providing significant insights into clinical diagnosis and management, particularly those examining inflammatory and thrombotic biomarkers, as well as imaging tools such as CT, MRI, and echocardiography. Emerging technologies and alternative biomarkers were also evaluated for their potential to improve diagnostic strategies. Special attention was given to biomarkers and imaging in genetic and autoimmune conditions like aortitis and connective tissue disorders linked to AAS. This review comprehensively examines evidence from experimental and clinical studies, offering a narrative overview of diagnostic strategies for AAS”. AI tools have been used in the preparation of the manuscript. The language corrections were made based on chatGPT.

## 3. Epidemiology and Risk Factors

The prevalence of aortic diseases is estimated at 1 to 3%, a seemingly high rate as it includes all aortic conditions [6]. The high mortality risk and the ambiguity surrounding the causes of out-of-hospital deaths make it challenging to accurately determine the prevalence of AAS. For example, nearly half of patients with IMH may progress to aortic dissection [7]. Hospitalizations for AAS have increased, likely due to age-related aortic deterioration, reduced wall strength under hypertension, and heightened vulnerability to atherosclerotic risk factors. According to a national analysis by Elbadawi et al. [8], involving 33,170 patients who underwent elective thoracic aortic aneurysm repair and were diagnosed with aneurysm-related acute aortic syndrome, hospitalizations for elective aortic repair increased by 1075 over four years (Ptrend = 0.01). Concurrently, hospitalizations for AAS rose by 795, with 780 of these cases classified as acute aortic dissection [8]. The epidemiological study by Xu et al. [9] corroborates this finding, reporting an average annual increase of 4%, ranging from 1 to 6%.

AAS is more common in men and is often associated with more comorbidities, but diagnosis and treatment are more challenging in women. The higher prevalence of cardiovascular diseases in men may limit physicians’ experience with female patients. Additionally, the distinct molecular structure and remodeling processes of the female aorta—characterized by higher levels of matrix metalloproteinase (MMP)-2 and MMP-9 and a more rapid decline in the expression of tissue inhibitors of metalloproteinases (TIMP)-1 and TIMP-2 over time—make female sex an independent risk factor for the development of AAS. Consequently, women face a higher risk of rupture, even in small aneurysms, and increased mortality rates despite treatment [10,11]. Data from the Canadian Thoracic Aortic Collaborative [12], which analyzed outcomes in patients undergoing arch surgery, identified female sex as an independent predictor of death (odds ratio [OR], 1.81; 95% confidence interval [CI], 1.24–1.89; *p* < 0.001), stroke (OR, 1.90; 95% CI, 1.28–2.85; *p* < 0.001), and adverse events (OR, 1.40; 95% CI, 1.16–1.69; *p* < 0.001). Similarly, a Swedish population-based cohort study examining patients undergoing planned proximal aortic surgery found that women are older, present with larger aortic diameters, and have an increased risk of 30-day mortality (3.1%) (OR, 2.25; 95% CI, 1.17–4.29). However, this association loses statistical significance after propensity score matching (OR, 1.31; 95% CI, 0.57–3.10) [13].

Any factor that increases stress on the aortic wall or disrupts its layers constitutes a risk factor for the development of aortic diseases. According to the International Registry of Acute Aortic Dissection (IRAD), uncontrolled hypertension and atherosclerosis are the primary etiological factors [14]. These risk factors are particularly significant in older patients; however, in younger individuals, genetic factors and collagen tissue disorders play a more prominent role [15].

Traumatic (Figure 1A–D) and iatrogenic (Figure 2A–D) aortic diseases, on the other hand, affect all age groups and require heightened clinical awareness [14,15]. Traffic accidents, which account for over one million deaths annually, according to World Health Organization (WHO) data, contribute significantly to traumatic aortic injury. Notably, one-fifth of these fatalities result from traumatic aortic injury, with the most commonly affected sites being the aortic isthmus (90%), aortic root (5%), and diaphragmatic hiatus (5%) [16,17]. Physicians should therefore maintain a high level of vigilance when evaluating patients in this context. 

## 4. Pathologic Definitions

### 4.1. Dissection, Intramural Hematoma, and Penetrating Aortic Ulcer

Understanding aortic anatomy is essential for diagnosing AAS. The aorta consists of the root, ascending aorta, arch, descending thoracic aorta, and abdominal aorta, each with distinct structural and functional features. Due to this segmentation, patients with AAS are typically classified based on whether the lesion involves the ascending aorta (type A; 59–67%) or does not (type B; 31%). Additionally, a separate category—referred to as non-A, non-B (3–10%)—includes cases where the lesion is confined to the aortic arch or extends retrogradely from a tear in the descending aorta without involving the ascending aorta. Among these conditions, classical dissection (CD) is the most common, while incomplete dissection (ID) accounts for approximately 5%. Intramural hematoma (IMH) and penetrating aortic ulcer (PAU) more frequently lead to type B lesions and are typically observed in older patients [18]. However, the exact prevalence rates are challenging to determine due to the overlapping nature of these conditions and their potential progression into one another [7].

Classical dissection (CD) involves the creation of a true lumen and a false lumen as blood separates the media layer following an intimal–medial tear (Figure 3A–D).

This process involves extracellular matrix disruption and elastic fiber fragmentation in the medial layer. Intimal tears typically occur in the right lateral wall of the ascending aorta or the descending segment near the left subclavian artery, areas prone to hemodynamic stress [19]. Clinically, CD typically presents with sudden and severe chest or back pain, often described by patients as “sharp and tearing”. CD has its own specific classification, and recently, the European update of the Stanford classification—the Type, Entry, and Malperfusion (TEM) classification—has been recommended for use. The full details are explained in Table 1, and the representation is shown in Figure 4 (Table 1 and Figure 4).

Incomplete dissection (ID) is defined by an intimomedial tear without the formation of a false lumen or an IMH. This absence of a false lumen often leads to diagnostic challenges [20] (Figure 5A–C).

Intramural hematoma (IMH) results from bleeding within the media, originating from the vasa vasorum, and is often associated with hypertension. It is traditionally defined as an aortic pathology lacking an intimomedial tear [21]. Uchida et al. [22] proposed that IMH should be reclassified as “thrombosed-type acute aortic dissection”, as they identified intimal tears in IMH patients either intraoperatively or through computed tomography imaging. The endothelial damage caused by atherosclerosis, along with the strong association between IMH and atherosclerotic lesions, suggests that IMH does not result from a single causative factor [23] (Figure 6A–C).

Another aortic pathology involving the intimal layer and closely associated with atherosclerosis is penetrating aortic ulcer (PAU). In this condition, the ulcer extends from the internal elastic lamina into the media and can progress to life-threatening complications, including aneurysm, dissection, or rupture [24] (Figure 7A–D).

### 4.2. Genetic-Associated Aortic Syndromes

At least one-fifth of thoracic aortic aneurysms (TAD) exhibit a hereditary pattern and are classified as hereditary thoracic aortic disease (HTAD). Mutations in genes associated with HTAD, which play key roles in aortic smooth muscle physiology, extracellular matrix regulation, and transforming growth factor-beta (TGF-β) signaling, are typically inherited in an autosomal dominant manner, although they may also be inherited in autosomal recessive or X-linked patterns. Patients with HTAD are generally younger (although older than those with Marfan syndrome), and their aneurysms are typically located in the ascending aorta, demonstrating significantly faster growth rates (an annual increase of 0.21 cm) [25] (Figure 8A–D).

Thijssen et al. [26] demonstrated a significant correlation between levels of matrix metalloproteinase-3 (MMP-3) and Insulin-like Growth Factor Binding Protein 2 (IGFBP-2) and aortic diameter in a cohort of 158 patients with TAD. Additionally, they identified elevated levels of Trem-like Transcript Protein 2 (TLT-2) in patients with hereditary TAD, suggesting its potential as a biomarker for genetic predisposition. Given its role in promoting leukocyte activation during inflammatory responses, TLT-2 expression may be linked to the inflammatory processes involved in aneurysm pathophysiology.

Genetic testing and biomarkers are crucial in family screenings for young patients with aortic root or ascending aortic aneurysms, as they help identify asymptomatic aneurysms in relatives. This approach allows for the determination of whether HTAD is familial or syndromic in nature. Marfan syndrome (FBN1; prevalence of 1/5000–1/10,000), Loeys–Dietz syndrome (TGFBR1, TGFBR2, SMAD3), and vascular Ehlers–Danlos syndrome (COL3A1) are the most well-recognized syndromic forms of HTAD, each associated with specific genetic mutations and distinct clinical features [27,28]. A diseased aorta typically enlarges silently over the years, with imaging techniques providing only a snapshot of its condition at a given moment. Therefore, it is crucial to fully understand the implications of these diagnostic tests.

Bicuspid aortic valve (BAV) pathology, observed in 5–20 per 1000 live births, is one of the most common congenital valve defects. It tends to exhibit a genetic predisposition and is frequently associated with aortopathy. While genetic testing is not recommended for isolated forms of BAV, the importance of thoroughly assessing family history has been emphasized [6]. In the presence of BAV, comprehensive imaging of the entire aorta is advised, and in cases of root or ascending aortic aneurysms, detailed evaluation of the valve pathology is essential (Figure 9A–D).

### 4.3. Autoimmune Aortitis

Inflammation of the aortic wall, regardless of whether it arises from infectious or non-infectious causes, is referred to as “aortitis”. Aortitis may remain asymptomatic for extended periods, present with a diverse spectrum of clinical manifestations, or progress to life-threatening complications such as dissection or rupture. Moreover, its classification within the broader category of “vasculitides” complicates efforts to accurately estimate its true prevalence [29]. Among non-infectious etiologies, giant cell arteritis (GCA; also known as Horton’s disease, commonly observed in women over 50 years of age) and Takayasu arteritis (TA; typically affecting women under 40) are the most frequent contributors to aortitis. GCA primarily affects medium-sized arteries, whereas TA predominantly affects elastic arteries, such as the aorta. As a result, these conditions should be considered in the differential diagnosis, particularly in younger patients presenting with recurrent syncope, cerebrovascular events, sudden vision loss or amaurosis fugax, or upper extremity circulatory disturbances (Figure 10A–C).

In GCA, vascular dendritic cells near the media layer trigger inflammation via sustained release of interferon-gamma (IFN-γ) and interleukin-17 (IL-17), leading to intimal disruption and smooth muscle cell damage. Elevated interleukin-6 (IL-6) and MMP-3 further exacerbate this process, causing progressive aortic wall injury [30]. TA is characterized by immune reactions similar to those seen in GCA, including the formation of necrotizing granulomas that lead to significant intimal thickening. These processes can result in various vascular complications, ranging from arterial stenosis to aneurysm formation. Mwipatayi et al. [31] underscore the severity of TA through a comprehensive 50-year retrospective analysis encompassing data from 272 patients. In addition, aortitis can present as isolated aortitis, typically confined to the thoracic aorta, or as part of immune-mediated inflammatory diseases such as rheumatoid arthritis, spondyloarthropathies, Behçet’s disease, Cogan syndrome, IgG4-related disease, relapsing polychondritis, systemic lupus erythematosus, and sarcoidosis, or as a consequence of radiation exposure [32,33] (Figure 11A–D).

The gold standard for diagnosing aortitis relies on biopsy results obtained from the temporal artery, lesion sites, skin, affected organs, or surgical specimens, combined with the expertise of an experienced pathologist. Currently, there is no definitive biomarker available for diagnosing aortitis without pathological methods. Nevertheless, elevated levels of CRP, erythrocyte sedimentation rate, smooth muscle myosin, serum amyloid A, cytokines—particularly IL-6—N-terminal pro-brain natriuretic peptide (NT-proBNP), pentraxin-3 (PTX3), and MMPs (especially MMP-2, MMP-3, and MMP-9), should be considered in the diagnostic workup [34].

In cases of clinical suspicion, imaging modalities play a pivotal role in diagnosing aortitis and guiding treatment decisions. Key tools in this process include echocardiography, CT, MRI, and positron emission tomography (PET). However, diagnosing aortitis remains challenging and necessitates a multidisciplinary approach involving close collaboration across clinical specialties [35].

## 5. Diagnosis

Goethe’s statement, “To learn to see is to learn the art of seeing”, aptly reflects the diagnostic process for aortic diseases. Clinical evaluation and imaging techniques provide the “act of looking”, while precise interpretation embodies the “art of seeing”. The complexity of aortic pathologies demands that clinicians move beyond superficial observation to gain a deeper understanding, facilitating early and accurate diagnosis. Although most patients with AAS present to the hospital with sharp chest pain, the symptoms can be misinterpreted due to the anatomical proximity of the aorta to neighboring structures. Furthermore, AAS is often referred to as a “masterful mimic”, as its complications frequently lead to diagnostic challenges [36]. For example, it may involve the aortic valve, causing acute decompensated heart failure due to aortic regurgitation; lead to pericardial tamponade from rupture; or result in mesenteric or renal ischemia through malperfusion. Additionally, an intimal–medial flap can obstruct the iliofemoral arteries, resulting in lower extremity ischemia; extend to the carotid arteries, causing cerebrovascular events; or affect the spinal cord, leading to paraplegia (Figure 12A–D).

New guidelines address diagnostic challenges by introducing supportive algorithms [6,37]. The 2024 ESC guidelines [6] start with the Aortic Dissection Detection Risk Score (ADD-RS), which is used when AAS is suspected. This score assigns one point for each category: high-risk conditions, pain characteristics, and examination findings. Based on the total score, the algorithm provides recommendations for further diagnostic evaluation.

Rogers et al. [38] developed a scoring system incorporating 12 clinical risk markers. According to this system, the most challenging patient group comprised 108 individuals who received a score of zero, as a score of zero does not definitively exclude AAS. Given the susceptibility of the aorta to various risks, it is plausible that future guidelines may adapt scoring parameters. For instance, the inclusion of additional risk factors—such as resistant hypertension, smoking or cocaine use, trauma, or a history of infection—could potentially alter the risk classification of patients who previously received a score of zero [39].

Moreover, refining the assessment of “pain” by scoring it under specific subcategories—such as location of spread, severity on a standardized pain scale, and time of onset—could make the evaluation more objective. This approach would enhance the clinical process and help physicians focus more effectively on the characteristics of pain, thereby improving diagnostic accuracy.

### 5.1. Laboratory Testing

There is no laboratory test specific to AAS that provides a definitive diagnosis; however, laboratory findings can be highly valuable in identifying complications of AAS or aiding in the differential diagnosis. For instance, the risk of rupture or bleeding can be assessed through hemoglobin monitoring, kidney function can be evaluated using creatinine levels, and liver function can be assessed via alanine aminotransferase (ALT) and aspartate transaminase (AST). Troponin levels are particularly useful in distinguishing myocardial infarction [40]. However, Vagnarelli et al. [41] emphasized the significance of troponin positivity in patients with AAS. They observed that in patients presenting with troponin positivity and electrocardiographic findings mimicking acute coronary syndrome (ACS), the diagnosis of AAS was delayed by an average of 3.5 h.

In acute inflammatory events, markers like C-reactive protein (CRP), erythrocyte sedimentation rate (ESR), and procalcitonin offer valuable insights. Combined with body temperature, white blood cell, and platelet counts, these markers help differentiate systemic inflammatory response syndrome (SIRS) and sepsis from AAS. Elevated platelet-to-lymphocyte ratio (PLR), neutrophil-to-lymphocyte ratio (NLR), and CRP levels have been established as significant predictors of mortality risk in patients with AAS [42,43]. A meta-analysis by Hsieh et al. [44] demonstrated that elevated CRP levels at admission are significantly associated with an increased risk of in-hospital mortality in patients with acute aortic dissection (HR = 1.15, 95% CI = 1.06–1.25, *p* = 0.001). While these tests provide useful insights, they are influenced by various diseases and represent only a minor step in the diagnosis of AAS. Consequently, the scientific community continues to explore more advanced diagnostic biomarkers [45] (Table 2). Although most of the tests listed in Table 2 are helpful in emergency situations, they primarily serve to predict mortality, monitor disease progression, or assist in screening efforts.

Among these biomarkers, D-dimer, which will be discussed in a separate section, has gained greater prominence in the literature [68,69].

#### 5.1.1. D-Dimer

D-dimer (DD), a fibrin degradation product elevated in fibrinolytic conditions like infarction and pulmonary embolism, is a key thrombotic biomarker for AAS. The role of DD in the diagnosis of AAS began to gain significant attention in the early 2000s. Erbel et al. [70], on behalf of the Task Force on Aortic Dissection, recommended the measurement of DD as a Class I/C indication in their guidelines. Weber et al. [71], in a prospective study, demonstrated that DD levels were significantly elevated in patients with acute aortic dissection, with a mean value of 9.4 µg/mL (range: 0.63–54.7 µg/mL). Furthermore, they found that this value doubled in dissections involving the entire aorta. Eggebrecht et al. [72] reported that, among patients grouped by the etiology of chest pain, DD levels provided an optimal cutoff value of 626 μg/L for diagnosing acute aortic dissection, with a sensitivity of 100% and a specificity of 73%. Similarly, Suzuki et al. [73], in a prospective multicenter study involving 220 patients, demonstrated that the commonly used cutoff value of 500 ng/mL for excluding pulmonary embolism could also effectively exclude acute aortic dissection within the first 24 h of presentation, yielding a sensitivity of 97% and a specificity of 59%. A 2021 meta-analysis further confirmed the diagnostic utility of DD, reporting a high pooled sensitivity (0.96) and moderate specificity (0.70), reinforcing its reliability for detecting true positives and ruling out disease (negative likelihood ratio: 0.06). Additionally, the diagnostic accuracy was supported by an area under the curve (AUC) of 0.94 [74].

Due to the elevation of DD levels in various conditions, the current guidelines recommend using DD in combination with the ADD-RS [6]. This combination is critical, as DD levels can be influenced by factors such as age, infection, pregnancy, trauma, and smoking [75]. A multicenter prospective study by Nazerian et al. [76] provides a significant contribution to this field. Based on data from 1850 patients, their findings indicate that a negative DD result in patients with an ADD-RS score of 0 or 1 results in a misdiagnosis rate of only 0.3 per 1000 cases, significantly reducing the need for radiological imaging. However, in patients with an ADD-RS > 1, advanced diagnostic investigations are recommended even when DD results are negative [76]. These findings are further corroborated by recently published meta-analyses [77,78].

DD levels are effective not only in the diagnosis of AAS but also in assessing disease severity and predicting in-hospital mortality rates. According to Itagaki et al. [79], patients with partial thrombosis in the false lumen exhibited the highest DD levels. Elevated DD levels (>8.3 μg/mL) have been identified as predictive of in-hospital mortality, particularly in the presence of complications such as shock, obesity, and malperfusion. Similarly, Zhao et al. [80] demonstrated that dissection reached its maximum transverse and longitudinal extent at DD levels equivalent to 2319 ng/mL or a natural logarithmic value of 7.749 (lnD-dimer). Their findings revealed a linear relationship between elevated DD levels and major adverse events, although no significant correlation with in-hospital mortality was observed. In contrast, Feng et al. [81] reported that DD levels exceeding 10,000 ng/mL were independently associated with mortality (OR: 3.17, 95% CI: 1.32–7.63, *p* = 0.010). Furthermore, elevated DD values were linked to a 12% increase in acute aortic dissection-related in-hospital mortality risk, even after adjusting for demographics and comorbidities (adjusted effect size: 1.12, 95% CI: 1.05–1.19, *p* < 0.01) [82].

#### 5.1.2. Homocysteine

Homocysteine (Hcy), a sulfur-containing amino acid, contributes to a strong atherogenic effect by inducing endothelial dysfunction, promoting vascular smooth muscle cell (VSMC) proliferation, and exposing the subendothelial matrix. These processes lead to platelet activation and extracellular matrix (ECM) remodeling. Hcy has been shown to promote elastolysis in the arterial media through a reactive oxygen species (ROS)-dependent process, either by activating MMP-2 or by directly targeting structural proteins such as fibrillin-1 and collagen [83]. Lee et al. [84] demonstrated that Hcy application to vascular endothelial cells (VEC) and VSMC, isolated from porcine thoracic aorta, significantly reduced VEC viability while simultaneously promoting VSMC proliferation. Through these effects, elevated Hcy levels contribute to the degradation of the aortic wall, potentially leading to the development of aneurysms or dissections. In a prospective study, Sbarouni et al. [47] compared patients with acute aortic dissection (AAD), ascending aortic aneurysms, and healthy individuals, demonstrating that while patients with aneurysms were older, Hcy levels were significantly higher in patients with AAD. However, Hcy levels in patients with AAD did not exceed 100 μmol/L. Smith and Refsum [85] identified at least 100 diseases associated with Hcy levels exceeding 11 μmol/L, with aneurysms being one of them. While the literature extensively discusses the relationship between Hcy and abdominal aortic aneurysms, the role of Hcy in AAD warrants more detailed investigation [86].

#### 5.1.3. Brain Natriuretic Peptide

Cardiac natriuretic peptides are a family of peptide hormones secreted by the atria and ventricles in response to increased wall tension. Although influenced by sex and age, cardiac natriuretic peptides are best known for their role in diagnosing ventricular hypertrophy and heart failure [87]. The strong association of AAS with left ventricular dysfunction and hypertension underscores the potential diagnostic value of BNP in aortic diseases. Sbarouni et al. [88] demonstrated that BNP levels were at least 20 times higher in aortic dissection compared to the healthy group (667 pg/mL vs. 31.3 pg/mL). However, they noted that DD is more useful in distinguishing between acute and chronic aortic pathologies. Although its diagnostic impact is moderate, a six-year prospective study demonstrated that NT-proBNP is a strong predictor of 30-day mortality (OR: 11.67, 95% CI: 2.61–52.09; *p* = 0.001) and major adverse events (OR: 50.21, 95% CI: 10.85–232.45; *p* < 0.001) in patients with aortic dissection who underwent surgery [89].

#### 5.1.4. Other Novel Potential Biomarkers

Beyond the biomarkers routinely tested in hospital laboratories, biomarkers identified through experimental or clinical studies using specialized test kits are presented, aiming to improve the diagnostic accuracy of AAS and predict its prognosis. Examples include creatine kinase-BB isozyme, MMPs, endothelin (a potent vasoconstrictor substance), smooth muscle myosin heavy chain (a major component of the smooth muscles), calponin (smooth muscle troponin-like protein), fibrin degradation products, angiotensin-converting enzyme, aggrecan (a proteoglycan found in cartilage), soluble elastin fragments (sELAF; elastin degradation product), tenascin-C, soluble suppression of tumorigenesis-2 (sST2), angiopoietin-like protein 8 (regulation of lipid metabolism and inflammation), plasminogen activator inhibitor 1 (PAI1), and interleukins [90] (Table 2). Forrer et al. [91] demonstrated that in the evaluation of chest pain etiology, Interleukin 10 (IL-10) showed a sensitivity of 55% and a specificity of 98% (with the cutoff value of 20 ng/L) for diagnosing acute aortic dissection (AAD). A combined model incorporating DD, high-sensitivity troponin T (hs-TnT), IL-6, and PAI-1 further improved diagnostic accuracy, achieving a sensitivity of 83% and specificity of 95%, correctly classifying 75% of AAD cases and 77% of all chest pain cases analyzed. Another study demonstrated that biomarkers associated with the smooth muscle structure of the aortic wall, including α-smooth muscle actin (α-SMA), smooth muscle myosin heavy chain, sELAF, and Polycystin 1 (which preserve the structural and functional stability of the vessel wall), were elevated in AAS and exhibited significant diagnostic value, especially when combined with DD [92]. Interleukin measurements could be more widely implemented, similar to DD testing, as hospital laboratories with active internal medicine and infectious disease clinics are often equipped for such analyses. As interleukins play a central role in the immune system, both physiologically and in pathological processes, they are found to be significantly elevated in AAS [93]. Particularly, IL-6 is recognized not only for its diagnostic utility but also as a prognostic marker in aneurysm patients treated with stent grafts. At levels exceeding the cutoff value of 18.36 pg/mL, IL-6 demonstrates a sensitivity of 87.4% and a specificity of 70.8% in predicting mortality [94].

### 5.2. Imaging

A detailed comparison of imaging modalities for aortic syndrome is presented in Table 3.

#### 5.2.1. X-Ray and Electrocardiography

Supported by clinical suspicion and laboratory tests, chest X-ray (CXR) and electrocardiography (ECG) are essential components in the second step of AAS diagnosis. While these tools are primarily utilized to rule out other potential causes of chest pain, such as pneumonia, pneumothorax, pleural effusion, or acute coronary syndrome, they also provide critical initial insights into aortic pathology [4,95]. Chest X-ray, with 64% sensitivity and 86% specificity for aortic disease diagnosis, is useful for detecting mediastinal widening, aortic contour enlargement, aortic knob dilation, and hemothorax [96,97] (Figure 13A–C).

#### 5.2.2. Echocardiography

Echocardiography (ECHO), through both transthoracic (TTE) and transesophageal (TEE) approaches, provides valuable insights into the evaluation of the proximal and distal segments of the aorta. ECHO plays a critical role in measuring aortic root diameters, assessing aortic valve pathologies, evaluating ventricular contractile functions, and detecting the presence of an intimal flap or pericardial effusion at an early stage (Figure 14A–C). Moreover, it particularly facilitates decision-making algorithms in patients with low ADD-RS [6].

Standard TTE exhibits a sensitivity of 85.35% and a specificity of 84.51% in detecting type A aortic dissection. These diagnostic rates improve to over 90% with the addition of contrast enhancement. Conversely, TEE achieves sensitivity and specificity exceeding 93% (93.6–99.87%) in evaluating both type A and type B dissections [98]. Wong et al. [99] corroborated these findings by detecting thoracic aortic aneurysms in 115 out of 1529 patients presenting to a hypertension clinic using Pocket-Size Mobile Echo. Echocardiographic evaluations are detailed in the guidelines; however, it is essential to emphasize that measurements can be influenced by factors such as chest deformities, patient noncompliance, body habitus, obesity, emphysema, rapid heart rate, false lumen jet flow velocity, and spiral flow within the aorta [100]. Furthermore, when using TTE, variations in aortic diameter depending on the systolic and diastolic phases should be considered. In TEE measurements, the potential impact of tracheal air must also be taken into account.

#### 5.2.3. Computed Tomography

The use of computed tomography (CT) and computed tomography angiography (CTA) is essential in the assessment of patients presenting with chest pain, particularly in emergency settings. The widespread availability of CT in most hospitals, along with its rapid results, repeatability, and ability to assist in diagnosing aortic diseases, risk assessment, treatment planning, and providing detailed information on both the lumen and wall structure of the entire aorta, justifies its significant role (Figure 15A–D). Additionally, non-contrast phase imaging allows for the estimation of aortic diameter, identification of aortic calcifications, detection of IMH, and evaluation of aortic stent conditions.

The contrast-enhanced phase is crucial, as it allows for the differentiation between the true and false lumens, identifies the rupture site, and determines the cause of obstruction in the aortic branches, whether due to flap-related or atherosclerotic/thrombotic occlusion (Figure 16A,B). The venous (late) phase aids in distinguishing flow within the false lumen from slow circulation and is useful in detecting endoleaks within stents [101] (Figure 17A,B). To minimize motion artifacts, particularly hyper-attenuation around the aortic wall, and to assess coronary arteries, it is recommended to acquire both thin-slice (<1.25 mm) and thick-slice (5 mm) images using electrocardiogram (ECG) gating [102] (Figure 18A,B).

Despite the 100% sensitivity and 98% specificity of CT in diagnosing AAS, initial misdiagnosis rates in emergency departments can reach up to 40% [103]. To mitigate this issue, it is crucial not only to recognize diagnostic pitfalls but also to adapt and expand the application of artificial intelligence systems, which have gained significant momentum across various fields, for the diagnosis of aortic diseases [104]. Hu et al. [105] reported an area under the receiver operating characteristic (ROC) curve (AUC) of 0.958 for DeepAAS, a deep learning-based artificial intelligence model designed to retrospectively analyze non-contrast CT images for the rapid and accurate diagnosis of acute aortic syndromes (AAS). This study also demonstrated that DeepAAS correctly identified 109 AAS cases initially misclassified as suspicious, achieving a sensitivity of 92.6% and a specificity of 99.2%. The model reduced the misdiagnosis rate from 48.8 to 4.8%, significantly expedited the diagnostic process, and shortened the average time to diagnosis by up to tenfold.

#### 5.2.4. Magnetic Resonance Imaging (MRI)

MRI offers distinct advantages in the diagnosis of AAS, primarily due to its ability to provide excellent soft tissue contrast without the risk of radiation exposure. It is particularly valuable in the evaluation of IMH, PAU, and other conditions where detailed visualization of the aortic wall. Advanced MRI techniques, such as phase-contrast imaging, allow for the assessment of blood flow dynamics, while non-contrast methods are particularly beneficial for patients with renal dysfunction. However, MRI has several limitations, such as its restricted availability, longer acquisition times, inapplicability in patients with metallic implants or hemodynamic instability, susceptibility to motion artifacts, and the potential to induce claustrophobia in awake patients. As a result, MRI is recommended for long-term follow-up in patients who have undergone aortic surgery or are under medical management to minimize renal damage and reduce radiation exposure.

## 6. Immuno-Metabolic Workup

Vascular inflammation, extracellular matrix degradation, apoptosis of VSMC in the medial layer, and endothelial dysfunction, all of which disrupt the cellular homogeneity of the aortic wall, play a critical role in the development of AAS. Following any trigger, the function of cells in the aortic wall becomes impaired or altered. Increased mechanical or oxidative stress on VSMC causes their transformation from a contractile phenotype to a reparative phenotype [106]. This process stimulates the migration of neutrophils, monocytes, and macrophages through the upregulation of vascular cell adhesion molecule-1 (VCAM-1) and intercellular adhesion molecule-1 (ICAM-1), ultimately compromising the structural integrity of the aortic wall. As a result, the DNA of VSMC undergoes degradation, initiating various forms of cell death [107]. Wang et al. [108] observed increased proteolytic activity (MMP-2), elevated caspase-3 activity indicative of apoptosis, and DNA fragmentation products in mice with angiotensin II-induced abdominal aortic aneurysms. Furthermore, reactive oxygen species (ROS) generated in cells under oxidative stress contribute to cell death through the activation of MMP-2 and MMP-9 [109]. In the experimental study by Li et al. [110], a significant number of macrophages were identified in the region of aortic dissection, accompanied by excessive MMP-9 expression at the tear site. The depletion of monocytes/macrophages has emerged as a potential strategy for preventing the development of aortic dissection.

The role of angiotensin II (AT-II) in aortic diseases has garnered increasing attention, particularly due to insights from preliminary studies. The renin–angiotensin system is well known for inducing vasoconstriction and hypertrophy, and AT-II contributes to an inflammatory cycle leading to cell death through the IL-6 cytokine pathway. Vasoconstriction results in hypertension, which disrupts diastolic blood flow in the vasa vasorum and triggers hypoxic stress in VSMC. AT-II exerts these effects both systemically and in a tissue-specific (local) manner. Experimental evidence showing that Losartan, an AT-II receptor antagonist, reduces aortic root growth by preventing aortic wall thickening, preserving elastic fibers, and attenuating transforming growth factor-beta (TGF-β) signaling in the aortic media underscores the effects of AT-II [111]. The atherogenic and inflammatory effects of AT-II are particularly evident in the context of apolipoprotein E deficiency.

The field of metabolomics has become an increasingly prominent and widely recognized discipline in aortic disease research, encompassing both endogenous metabolites, which are gene-protein products, and exogenous metabolites derived from dietary intake or medications [112]. Altered metabolite levels can play a significant role in the development or prevention of various diseases. These metabolites can trigger inflammation through carbohydrate, lipid, and amino acid metabolism in all three layers of the aorta, increase the release of ROS, exert vasoconstrictive effects, and activate the transition of VSMC to a synthetic phenotype [113]. Table 4 provides examples of the most studied and widely researched metabolites and their effects (Table 4).

## 7. Discussion

The EACTS/STS guideline begins by defining the aorta as “a self-sufficient part of an organism with a specific vital function” [37]. The fate of the aorta, described as a “heterogeneous functional unit” during the embryological period, makes this structure susceptible to various factors, including mechanical stress induced by blood flow and pressure, aging, environmental influences, endogenous and exogenous metabolites, genetic and connective tissue disorders, and infections. A healthy aortic wall is a dynamic and physiologically active structure that resists or adapts to these challenges and is composed of three fundamental, interacting layers. The intimal, medial, and adventitial layers of the aorta exhibit structural and functional heterogeneity across its segments, leading each segment to respond differently to hemodynamic influences [132].

In the experimental study by Giudici et al. [133], it was observed that the ascending aorta significantly increases the storage of elastic energy in response to both circumferential and axial loads, while the lower thoracic aorta shifts its load-bearing capacity from the elastic media to the stiff adventitia. This phenomenon helps explain why hypertension is a higher risk factor in type B aortic dissections compared to type A (81 vs. 74%) [134].

Atherosclerotic disease of the aorta is the most common aortic pathology, occurring in up to 50% of cases, followed by aortic dilation, aneurysm, and aortic syndromes. Despite its rarity, AAS carries a significant risk of high mortality. A recent analysis across 33 European countries demonstrated a minimum 35% increase in mortality rates due to aortic aneurysm and dissection over the past two decades [135]. The low incidence of AAS, estimated at 3–10 per 100,000, despite its high mortality, can be attributed to several factors. These include the fact that many patients die before reaching a hospital, misdiagnosis in those who receive medical care, or delays in obtaining an accurate diagnosis [136]. According to a 60-year autopsy review by Huynh et al. [137], at least 60% of patients with aortic dissection received their initial diagnosis post-mortem. The non-specific nature of chest pain contributes to misdiagnosis or delayed diagnosis in approximately one-third of patients, often due to its clinical overlap with conditions such as acute coronary syndrome or pulmonary embolism [138]. These high misdiagnosis rates have necessitated the development of diagnostic algorithms and the investigation of supportive biomarkers to improve the identification and management of the condition [6,37]. In dissection, sudden and sharp chest pain is the typical symptom; however, 2–7% of patients do not experience pain. Additionally, most aneurysms are discovered incidentally [14]. Symptoms originating from various body systems can involve the aorta to varying degrees. For instance, patients may present with symptoms such as dysphagia, hemoptysis, vomiting, melena, hoarseness, ptosis, or stroke, leading them to other specialties [139]. In an analysis of 1663 patients, Lovatt et al. [140] reported the overall rate of misdiagnosis for aortic dissection as 33.8% (ranging from 14.1 to 78.3%). This variability may be attributed to the inclusion of both retrospective and older studies in the analysis.

The diagnosis of AAS begins with clinical suspicion and is further guided by clinical expertise during patient assessment. A thorough examination of the patient’s symptoms, body structure, and overall condition may provide critical diagnostic insights. Findings from the physical examination are then used to calculate the risk score, which integrates clinical, historical, and examination data to stratify risk [6,37]. This risk score, in conjunction with laboratory results, ECG, and chest X-ray findings, helps determine the need for advanced imaging. For example, in patients with a low ADD-RS score, a DD level of ≥500 ng/mL indicates the necessity of a CT scan [139].

DD is the most commonly used biomarker in the diagnosis of AAS, widely accepted for both diagnosis and differential diagnosis. Its sensitivity and specificity vary based on cutoff values defined in different studies. Sakamoto et al. [141] reported significantly higher DD levels in patients with acute aortic dissection (32.9 ± 66.7 g/mL, *p* < 0.001) and pulmonary embolism (PE; 28.5 ± 23.6 g/mL, *p* < 0.001) compared to those with acute myocardial infarction (AMI; 2.1 ± 3.7 g/mL). A cutoff value of 5.0 g/mL was found to effectively differentiate dissection and PE from AMI, with a sensitivity of 68% and specificity of 90%. Kaito et al. [142] identified acute aortic dissection in 28 out of 322 patients diagnosed with ST-segment elevation myocardial infarction based on DD levels of ≥750 ng/mL. Elevated DD levels, reflecting thrombotic and fibrinolytic activity, are associated with a wide range of conditions. Following the COVID-19 pandemic, prolonged D-dimer elevation may occur as a response to inflammation, making accurate interpretation of elevated levels crucial [143]. Therefore, DD levels should be interpreted alongside the ADD-RS and other diagnostic modalities to ensure accurate clinical assessment. Although numerous biomarkers other than DD have been reported in the literature (Table 1), their limited clinical use has led to DD remaining the most widely used and established biomarker in practice.

In the diagnosis of AAS, CT is considered the gold standard; however, echocardiography (both transthoracic and transesophageal) and ultrasound (point-of-care and pocket-sized devices) are also effectively utilized. Pocket-sized ultrasonic devices are practical tools that should be accessible in all clinical settings. They are highly sensitive and specific for detecting intimal flaps (especially in type A dissections), arterial dilation, aneurysms, valve regurgitation, and pericardial or pleural effusion. Their primary advantage, however, is in expediting the diagnostic process, particularly in hemodynamically unstable patients [144]. The PROFUNDUS study [145] demonstrated that when ultrasound and DD were used together, the applied protocol reduced the need for CT imaging by 41%. This approach is expected to decrease the unnecessary use of CT, which, although highly effective in acute conditions, carries the risk of overuse.

CT should be performed promptly in high-risk cases, typical clinical presentations, or when echocardiography or ultrasound findings warrant it. However, its risks—contrast reactions, renal injury, and radiation exposure—must be considered [146]. Despite these risks, IRAD data show that the use of CT has doubled over the past 17 years [134]. While CT plays a central role in diagnosing AAS, its limitations stem from varying imaging presentations across different sequences and potential gaps in knowledge or failure to consult experienced cardiovascular specialists or radiologists [147]. Dreisbach et al. [148] highlighted the importance of expertise, noting that 31% of AAS cases had diagnostic discrepancies due to unrecognized or misinterpreted AAS, misclassification, or complications like hemorrhage. Artificial intelligence systems can enhance diagnostic accuracy and enable higher-quality imaging with reduced radiation exposure [149].

## 8. Future Directions 

The legal implications of misdiagnosis or failure to diagnose, as well as the societal impact of morbidity and mortality, underscore the urgent need for improved diagnostic protocols. Advanced clinician training and system-wide strategies are essential to address persistent gaps in AAS screening and the absence of validated biomarkers [150,151,152]. Risk assessment and screening programs can identify vulnerable aortas but are insufficient to reduce disease incidence or protect patients effectively. This is because an increased aortic diameter indicates medial degeneration, a marker of aortopathy, rather than the disease itself. Childhood studies focusing on triggers such as past infections, microbiota imbalances, obesity, vitamin D deficiency, and environmental factors, as well as analyzing the mechanical and hemodynamic behavior of the aorta, could help prevent catastrophic outcomes of AAS [153,154,155].

Although AAS diagnosis is more successful in high-volume centers or hospitals with dedicated aortic teams, biomarkers play an increasingly critical role in effective triage for aortic dissection, especially in primary or intermediate healthcare settings. However, biomarker levels generally begin to rise after damage to the aortic wall and vary depending on the timing of symptom onset. While DD remains the most prominent biomarker today, its levels exhibit significant variability across the AAS spectrum and are heavily influenced by demographic factors and a range of other medical conditions [45,156]. This underscores the need for standardized cutoff values and more comprehensive studies that integrate DD with routinely available inflammatory markers to enhance its diagnostic relevance and facilitate practical implementation [157]. Additionally, if the diagnostic power of other biomarkers is validated through larger-scale prospective studies, they could be incorporated into future clinical guidelines. Achieving standardized cutoff values for biomarkers will require international collaboration and large-scale multicenter studies.

Liquid biopsy techniques, such as the analysis of circulating extracellular vesicles and cell-free DNA, hold the potential for early detection of vascular injuries, including AAS [158]. Future research could explore their diagnostic and prognostic utility, particularly in combination with established biomarkers such as D-dimer. Furthermore, artificial intelligence-driven diagnostic algorithms and community-based screening initiatives could complement traditional methods, enabling earlier detection and personalized interventions. The development of portable and cost-effective imaging technologies could also enhance diagnostic capabilities in resource-limited settings. Imaging modalities will remain the primary diagnostic tool until a vascular-specific biomarker is identified. However, integrating advanced computational models simulating aortic hemodynamics with imaging techniques could provide insights into rupture risk and guide surgical planning. Additionally, emerging nanotechnology applications, such as nanoparticle-based biosensors and targeted contrast agents, offer promising avenues for enhancing biomarker detection and imaging precision in AAS diagnosis [159].

## 9. Conclusions

The heterogeneity of AAS presentations and the complexity of its underlying mechanisms continue to pose significant challenges in achieving accurate and timely diagnosis. Currently, the most reliable diagnostic tools are DD, which is widely available and demonstrates high specificity and sensitivity, and CT, which offers rapid and accessible imaging. These modalities have become cornerstones in the diagnostic workflow of AAS. Experimental models and patient studies have often taken a unidirectional approach by focusing on specific biomarker groups, with their diagnostic success being demonstrable only when combined with prospective studies and imaging modalities. However, it is essential to acknowledge the limitations of imaging tools, such as radiation exposure, and the time-consuming nature of techniques like MRI and PET. Future studies should focus on integrating genetic and epigenetic data to uncover the mechanisms of AAS while incorporating AI into clinical workflows to enhance imaging analysis and risk factor synthesis. These advancements are enhancing diagnostic accuracy, risk stratification, and patient outcomes through personalized insights and predictive modeling. Addressing diagnostic gaps requires implementing primary care screening programs, establishing specialized aortic teams in high-volume centers, and developing non-invasive imaging modalities. These advancements would enable timely diagnosis, provide real-time insights into aortic dynamics, and enhance early detection while minimizing risks associated with radiation and contrast agents.

## Figures and Tables

**Figure 1 diagnostics-15-00183-f001:**
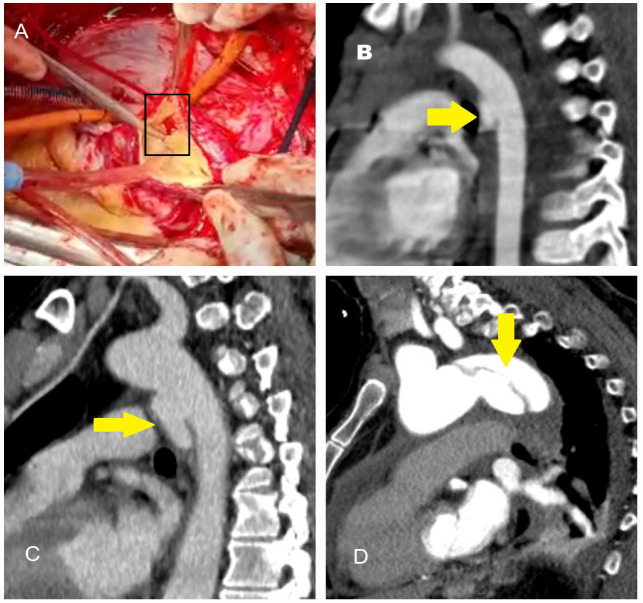
Operative and computed tomography angiography images illustrating traumatic aortic injury. (**A**) A 45-year-old male patient involved in a motor vehicle accident with chest impact against the steering wheel underwent surgery due to massive pericardial effusion detected on echocardiography. The rupture site was located on the right lateral wall of the ascending aorta (black box). (**B**) A 9-year-old male patient who fell from a walnut tree was found to have a transection just distal to the subclavian artery on computed tomography angiography performed in the emergency department. The injury was treated with a stent graft. (**C**) A 35-year-old male farmer was treated for descending aortic dissection following a tractor rollover accident. (**D**) A 63-year-old female patient was treated for descending aortic dissection resulting from a motor vehicle accident outside the vehicle (The yellow arrows indicate the location of the dissection).

**Figure 2 diagnostics-15-00183-f002:**
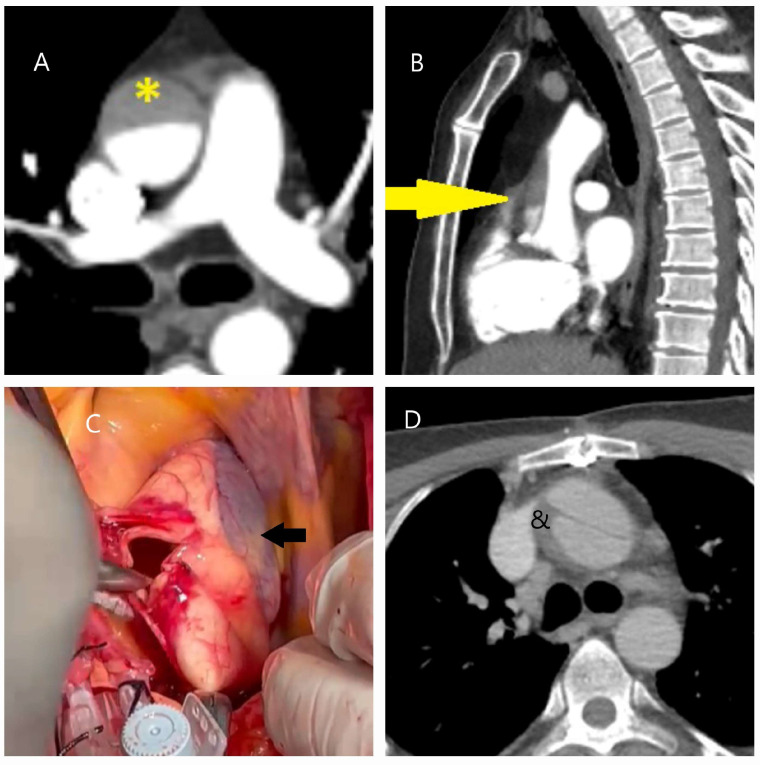
Computed tomography and operative images demonstrating iatrogenic aortic injury. (**A**,**B**) Axial (asterisk) and sagittal (yellow arrow) CT images showing dissection in the right anterolateral segment of the ascending aorta, which developed during imaging of the right coronary artery in a 50-year-old male patient with inferior myocardial infarction. (**C**) Angiographic image showing a hematoma in the right lateral wall of the ascending aorta (black arrow) that developed during angiography in a 60-year-old male patient. (**D**) CT image showing a hematoma and dissection at the aortic cannulation site (indicated by the ampersand symbol) in a 45-year-old female patient who had undergone surgery for bicuspid aortic valve one month prior to the dissection event.

**Figure 3 diagnostics-15-00183-f003:**
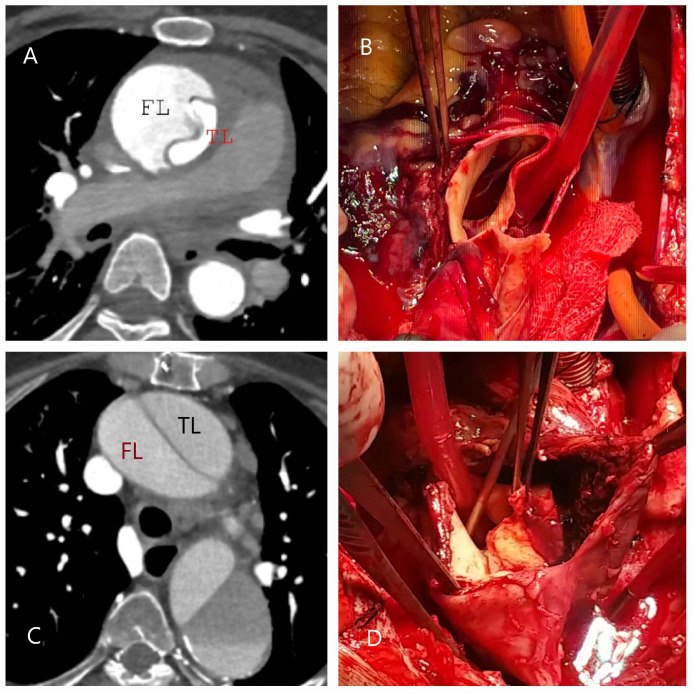
CT and operative images illustrating true lumen (TL) and false lumen (FL) in classic aortic dissection. (**A**,**B**) Images from a 48-year-old male long-haul truck driver with no known risk factors other than smoking. He was diagnosed with type A aortic dissection following severe, tearing chest pain triggered by emotional stress. In (**B**), the aspirator demonstrates the false lumen (FL). (**C**,**D**) Images from a 65-year-old female patient with diabetes mellitus and hypertension who was diagnosed with aortic dissection in the emergency department four days after persistent back pain radiating distally from between the scapulae. The pincette highlights the intimomedial flap, and the FL shows a thrombotic appearance (TL = true lumen; FL = false lumen).

**Figure 4 diagnostics-15-00183-f004:**
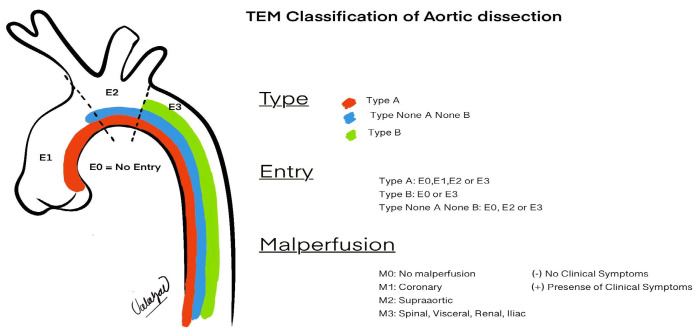
Schematic representation of the TEM classification (descriptions are provided in Table 1). Derived from the 2024 ESC Guidelines [6].

**Figure 5 diagnostics-15-00183-f005:**
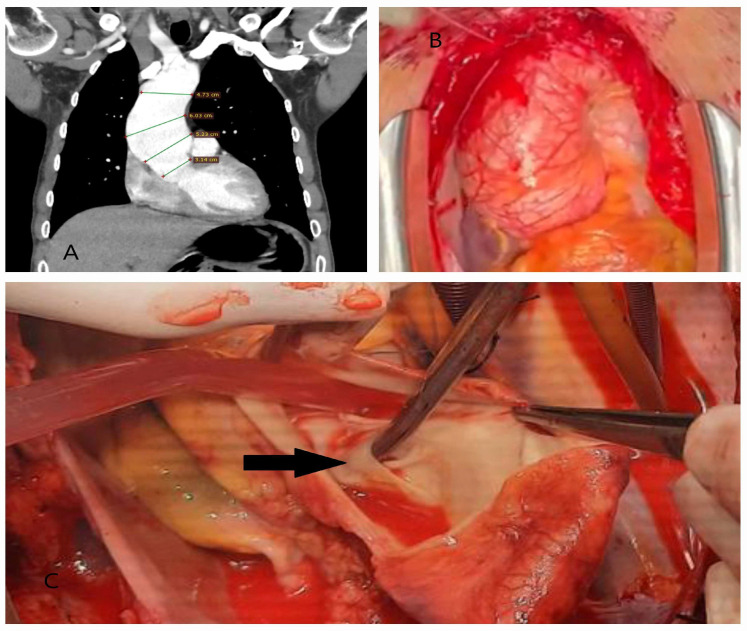
CT and operative images of incomplete dissection. A 28-year-old male farmer with a marfanoid physical appearance but no notable family history was evaluated for exertional dyspnea and diagnosed with severe aortic insufficiency. (**A**) Coronal CT image showing an ascending aortic aneurysm. (**B**) Operative image of the ascending aorta. Pay attention to the absence of findings indicative of aortic dissection in both the CT and operative images. (**C**) During exploration of the ascending aorta, which appeared solely as an aneurysm on imaging, a pocket-like, self-contained dissected area was identified (black arrow). Genetic testing for the patient is ongoing.

**Figure 6 diagnostics-15-00183-f006:**
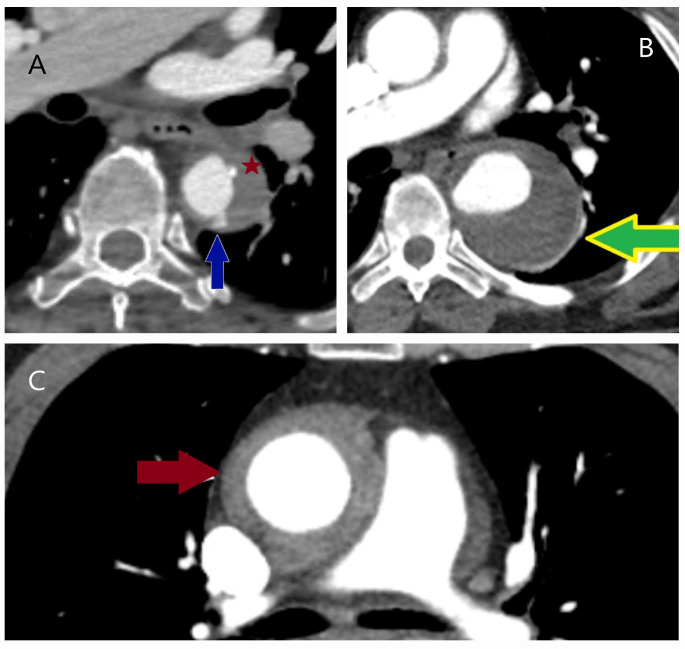
CT images of intramural hematoma and periaortitis. (**A**) Axial CT image of a 68-year-old patient with a history of hypertension, smoking, and coronary artery bypass grafting. The site of the intimal defect is visible (blue arrow), and intimal calcifications (typically associated with atherosclerotic risk factors) are distinguishable features (red star). (**B**) CT image of a 77-year-old male patient diagnosed with type B intramural hematoma detected during routine scans (green arrow) while undergoing treatment for chronic obstructive pulmonary disease. (**C**) CT image of a 58-year-old male patient who presented with intermittent low-grade fever and accompanying chest pain during the day. Echocardiography revealed thickening of the ascending aortic wall and an increase in diameter, prompting further evaluation with CT. The scan identified circular, non-luminal periaortic findings consistent with periaortitis. This condition, often associated with diseases such as Erdheim–Chester disease or IgG4-related vasculitis, requires further investigation to avoid misdiagnosis as intramural hematoma (red arrow).

**Figure 7 diagnostics-15-00183-f007:**
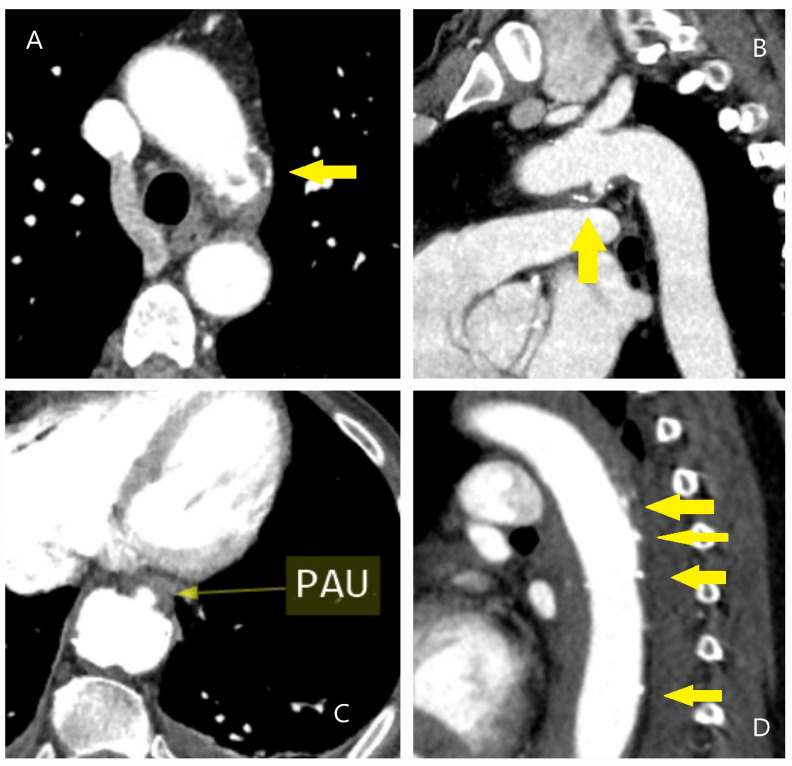
CT images of penetrating aortic ulcer (PAU). (**A**) CT image of a 72-year-old male patient undergoing coronary CT angiography, revealing an incidental finding of atheromatous plaque disruption with a penetrating ulcer extending into the intima within the aortic arch. (**B**) Sagittal CT image of the same patient, providing a clearer view of the aortic ulcer. This highlights the importance of evaluating axial images alongside other CT slices for accurate assessment. (**C**,**D**) CT images of a 65-year-old female patient with a history of cancer treatment, hyperlipidemia, hypothyroidism, and smoking. The scans, performed due to back pain, incidentally revealed a penetrating aortic ulcer. Yellow arrows indicate the PAU.

**Figure 8 diagnostics-15-00183-f008:**
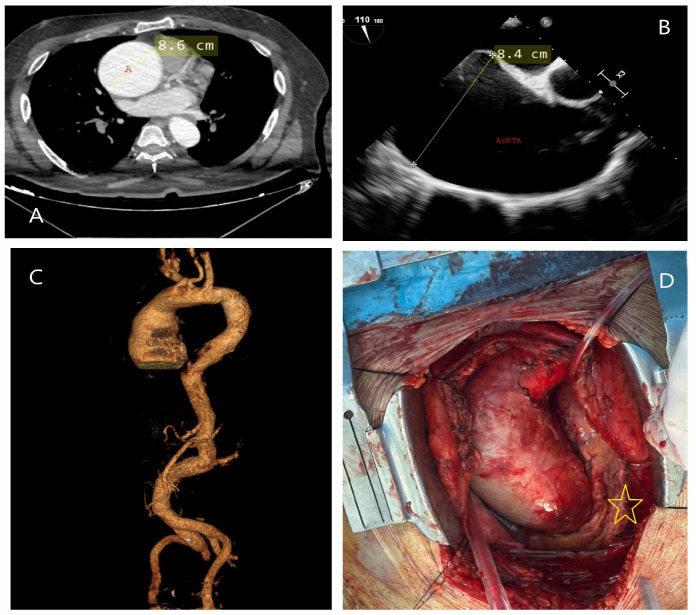
CT, TEE, and operative images of hereditary thoracic aortic aneurysm. A 58-year-old asymptomatic male patient, evaluated for screening purposes due to his brother’s history of aortic dissection, was incidentally found to have a giant ascending aortic aneurysm. (**A**) Axial CT image showing the aneurysm measuring 8.6 cm.; A, Ascending aorta (8.6 cm) (**B**) TEE performed to assess aortic valve pathology revealed a tricuspid valve with severe aortic insufficiency, and the ascending aorta measured 8.4 cm. (**C**) 3D CT angiography evaluating all segments of the aorta to investigate the presence of additional aneurysms. (**D**) Intraoperative image showing the ascending aorta expanding circularly to occupy the entire mediastinum and displacing the heart to the left (yellow star). The patient underwent a Bentall procedure, and ventricular functions improved postoperatively.

**Figure 9 diagnostics-15-00183-f009:**
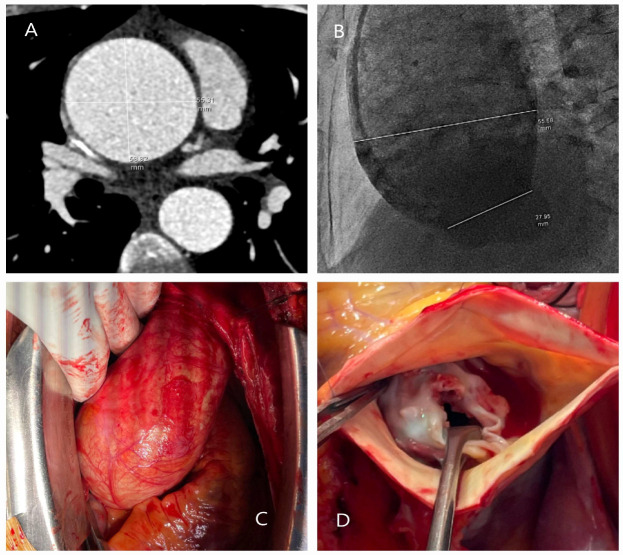
CT, aortography, and operative images of a patient with a bicuspid aortic valve causing an ascending aortic aneurysm. A 55-year-old male patient was under follow-up for known bicuspid aortic valve disease. Despite initially refusing surgery due to fear of the procedure, he decided to undergo surgery due to worsening dyspnea and decreased functional capacity in recent months. (**A**) Axial CT image showing the ascending aorta measuring over 50 mm. (**B**) Aortography confirming the diameter of the ascending aorta exceeds 50 mm. (**C**) Operative image showing the post-stenotic aneurysm with an enlarged aorta (in cases of root aneurysms or bulb-like aortic shapes, bicuspid aortic valve should always be investigated). (**D**) Intraoperative image demonstrating the calcified and fused cusps of the bicuspid aortic valve.

**Figure 10 diagnostics-15-00183-f010:**
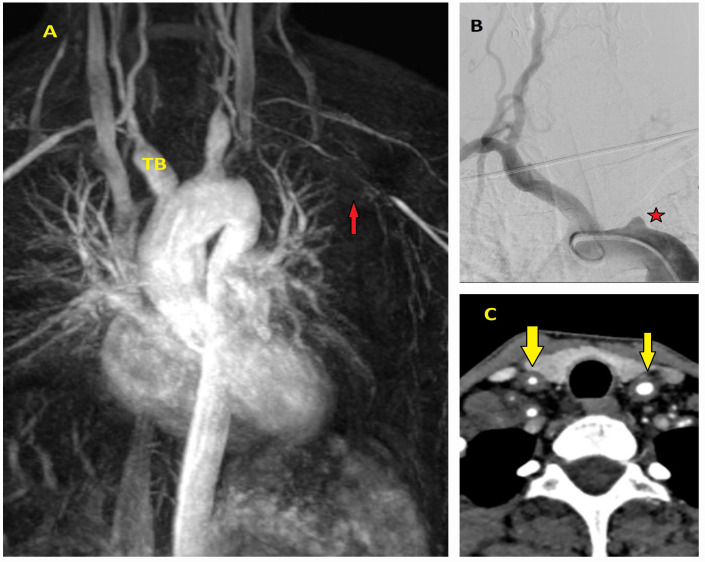
CT, MR, and angiographic imaging of Takayasu arteritis and giant cell arteritis. (**A**) A 28-year-old female patient presenting with symptoms of arm numbness and weakness during activity was initially evaluated for thoracic outlet syndrome and cervical discopathy. However, MR imaging revealed findings consistent with Takayasu arteritis, demonstrating long-segment caliber loss in the left subclavian and carotid arteries (red arrow; TB, Truncus Brachiocephalicus). (**B**) A 32-year-old male smoker presented with ischemia in the left upper extremity. Clinical history revealed frequent episodes of amaurosis fugax. Doppler ultrasonography of the carotid and upper extremities showed no detectable flow, and angiography confirmed occlusion (red star). The patient was diagnosed with Takayasu arteritis following a temporal artery biopsy. (**C**) A 62-year-old female patient from the Caucasus with a recent history of worsening headaches, dizziness triggered by sudden movements, persistent tinnitus, and mildly elevated D-dimer levels was initially suspected to have long COVID syndrome. However, Doppler ultrasound and subsequent CT imaging revealed fibrotic thickening of the arterial walls in both carotid arteries, with significant stenosis in the right carotid artery (yellow arrows). Temporal artery biopsy confirmed a diagnosis of giant cell arteritis.

**Figure 11 diagnostics-15-00183-f011:**
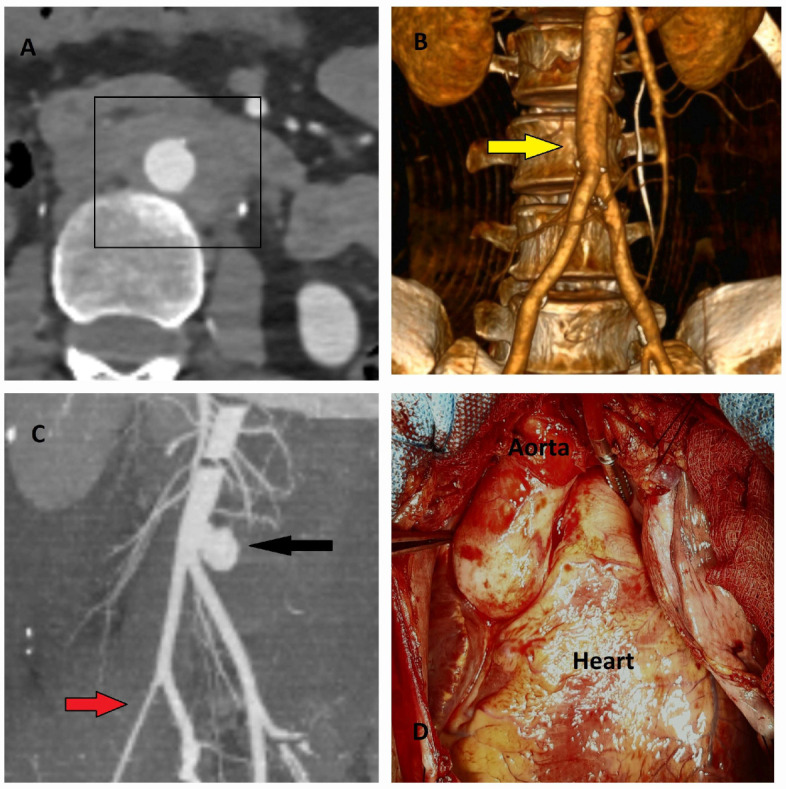
CT and surgical images of aortitis associated with immune-mediated inflammatory diseases. (**A**) A 58-year-old male farmer presented with weight loss and claudication. CT imaging revealed circular fibrotic tissue surrounding the abdominal aorta (black square) along with retroperitoneal fibrosis, leading to a diagnosis of Erdheim–Chester disease. (**B**) A 3D CT angiography of the same patient showed the aorta appearing normal, resembling that of a healthy individual (yellow arrow). These images emphasize the need for meticulous examination of CT findings, especially when evaluating aneurysms and intramural hematomas. (**C**) A 25-year-old male patient presented with recurrent pain in the lower right abdominal quadrant. He reported a history of frequent oral ulcers and occasional vision problems, suggestive of Behçet’s disease. Despite ongoing medication, the patient experienced persistent attacks. During his third-year follow-up, he was evaluated for claudication and was found to have a saccular aneurysm in the abdominal aorta (black arrow) and a diminutive right iliac artery (red arrow). (**D**) A 22-year-old male heavy smoker underwent early coronary artery bypass grafting due to acute coronary syndrome caused by a left main coronary lesion. Intraoperatively, the ascending aorta appeared small relative to his body size, irregularly shaped, and covered with a marble-white fibrous tissue. Biopsy findings confirmed a diagnosis of IgG4-related aortitis. He is currently under follow-up by the rheumatology clinic.

**Figure 12 diagnostics-15-00183-f012:**
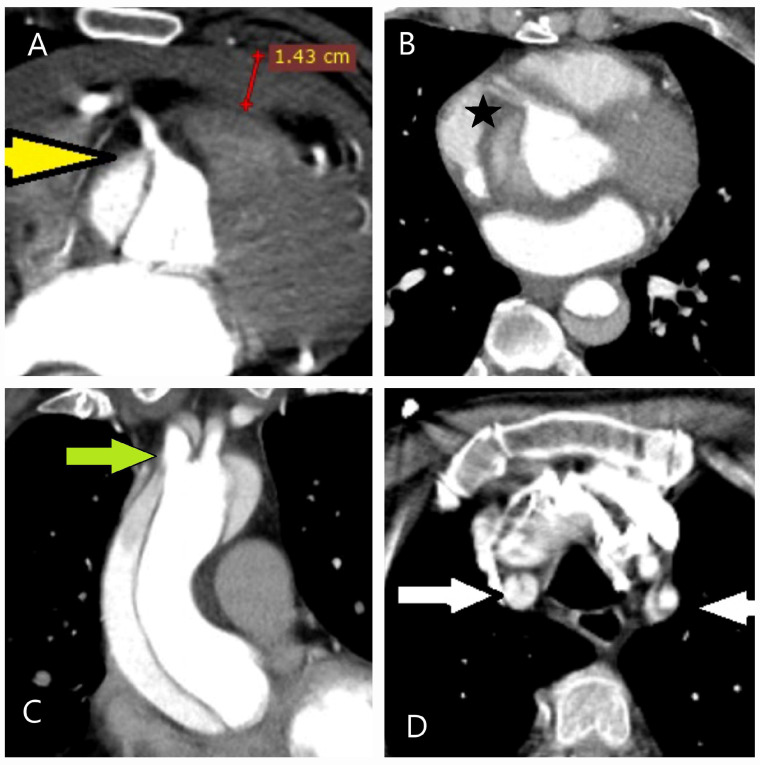
Vascular complications of aortic dissection. (**A**) CT image of a 52-year-old female patient who presented to the emergency department with tearing chest pain. Electrocardiography revealed findings consistent with inferior myocardial infarction, but elevated D-dimer and C-reactive protein levels prompted further evaluation with CT, leading to a diagnosis of aortic dissection. The dissection flap was found to originate from the right coronary artery ostium (yellow arrow). Additionally, 1.43 cm of pericardial effusion due to the dissection was observed. (**B**) CT image of a 62-year-old male patient presenting with findings of inferior myocardial infarction. Point-of-care echocardiography in the emergency department raised suspicion of an intimal flap. Further evaluation revealed that the type A aortic dissection also dissected the right coronary artery (black star). (**C**,**D**) CT images of a 65-year-old male patient on antihypertensive medication who presented to the emergency department after experiencing syncope following upper extremity numbness. The dissection was found to extend along the brachiocephalic trunk (green arrow) and carotid arteries (white arrows).

**Figure 13 diagnostics-15-00183-f013:**
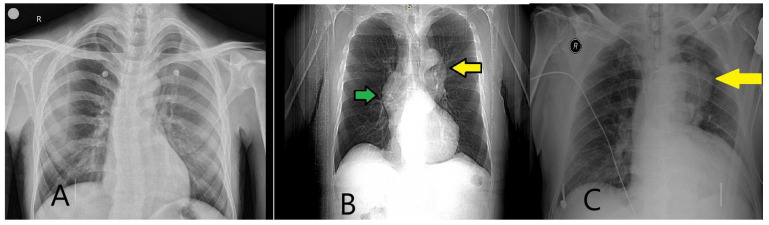
X-ray images associated with aortic diseases. (**A**) X-ray image of a 25-year-old male patient with Marfan syndrome, showing a long and narrow thoracic cage and downward-appearing heart. Spinal curvature is also observed. (**B**) X-ray image of a 62-year-old male patient showing projections in the mediastinum corresponding to the ascending aorta. The yellow arrow indicates the aortic knob, while the green arrow highlights the prominence of rightward enlargement. (**C**) X-ray image of a 75-year-old male patient showing a prominent aortic knob (yellow arrow). Further evaluation of these suspicious findings is recommended.

**Figure 14 diagnostics-15-00183-f014:**
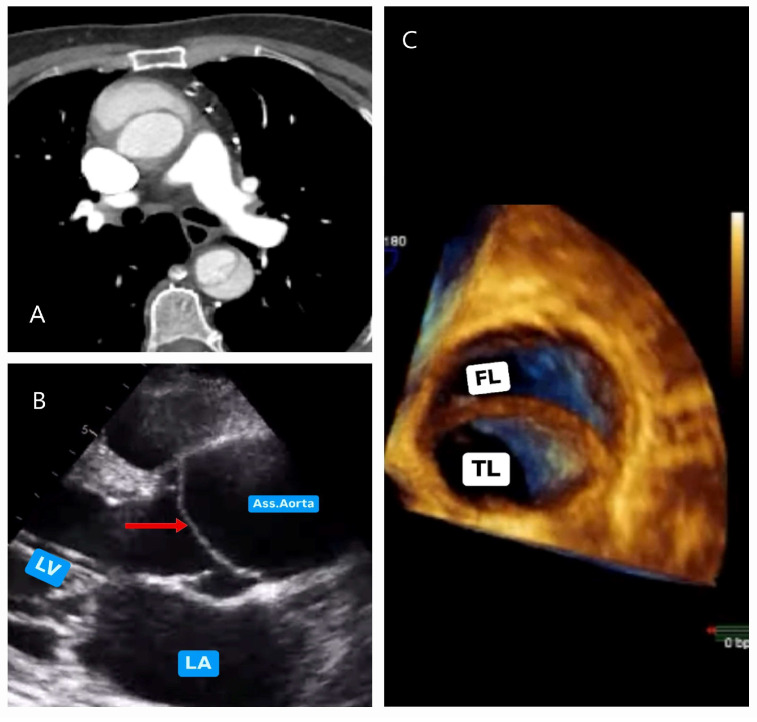
Echocardiographic images in acute aortic dissection. (**A**) CT image of a 52-year-old female patient evaluated in the emergency department, leading to the diagnosis of type A aortic dissection. (**B**) Point-of-care transthoracic parasternal long-axis echocardiography showing the intimal flap (red arrow), performed to assess the condition of the aortic valve. (**C**) Three-dimensional transesophageal echocardiography (TEE) image obtained after anesthesia induction, demonstrating the true lumen (TL) and false lumen (FL). Performing TEE for every patient undergoing surgery provides critical insights for valve repair planning (LA, left atrium; LV, left ventricle).

**Figure 15 diagnostics-15-00183-f015:**
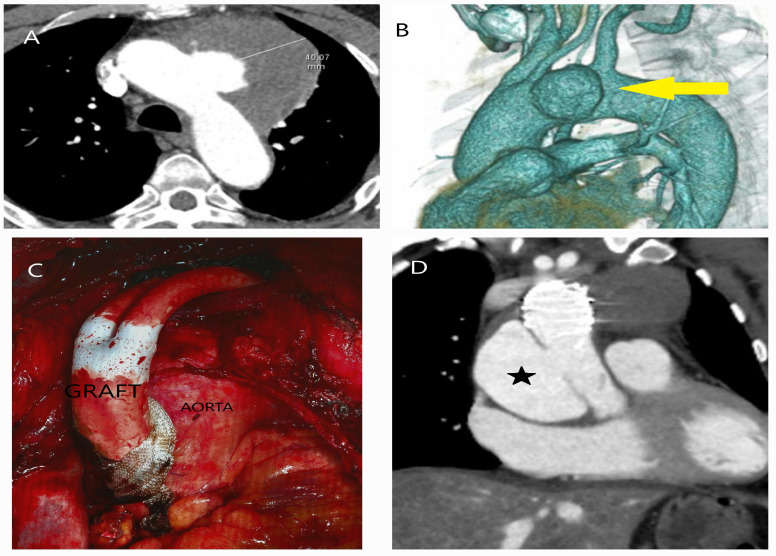
The role of CT in the evaluation of the aorta for treatment planning. A 52-year-old male of Caucasian descent was diagnosed with an aortic arch aneurysm after a vigilant gastroenterologist detected pulsations in the esophageal wall during endoscopy, performed to investigate the etiology of dysphagia. (**A**) Axial CT image showing the aneurysm in the aortic arch. (**B**) Translucent 3D CT image highlighting the aneurysm in the aortic arch (yellow arrow). (**C**) Based on the CT findings, the patient underwent a debranching operation with a bifurcated graft and simultaneous thoracic endovascular aneurysm repair (TEVAR). (**D**) Follow-up CT one year later revealed the development of an aneurysm and dissection at the graft anastomosis site in the ascending aorta (black star).

**Figure 16 diagnostics-15-00183-f016:**
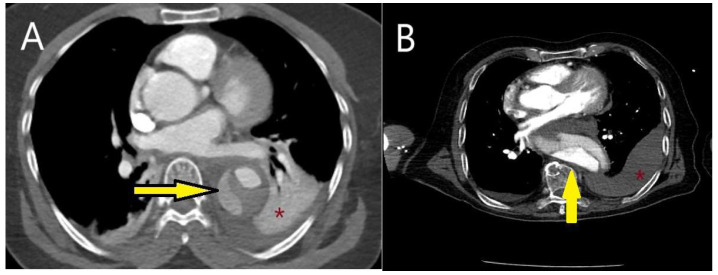
The use of CT in identifying rupture. (**A**) CT image of a 57-year-old female patient with a history of hypertension who presented to the emergency department with paraplegia and sudden-onset dyspnea. Hypotension prompted further evaluation with CT, which revealed a type B aortic dissection rupture into the left thoracic cavity. The yellow arrow indicates the “Yin-Yang phenomenon” within the descending aorta, suggesting continued flow in the false lumen. The patient was treated with emergency TEVAR under spinal cord pressure-reducing therapy; however, the paraplegia remained permanent. (**B**) CT image of an 80-year-old male patient diagnosed with type B acute aortic dissection, which led to hypotensive shock and rupture into the left thoracic cavity. * Red asterisk indicates pleural effusion.

**Figure 17 diagnostics-15-00183-f017:**
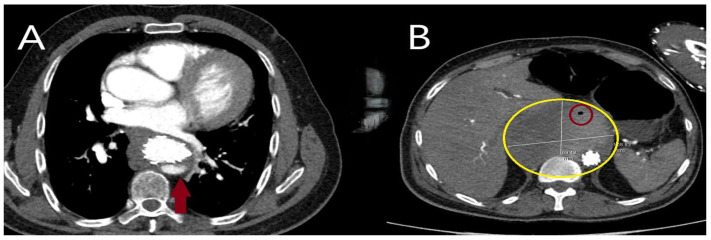
The use of CT in detecting stent and stent-related complications. (**A**) CT image of a 75-year-old male patient with chronic obstructive pulmonary disease (COPD), showing an endoleak into the aneurysm during follow-up of the TEVAR stent (red arrow). (**B**) CT image of a 50-year-old male patient with achondroplasia undergoing stent follow-up for a thoracic aortic aneurysm. The area within the yellow circle indicates the aneurysm, while the red circle highlights the presence of air within the aneurysm. If air bubbles increase during follow-up, stent or aneurysm infection should be considered.

**Figure 18 diagnostics-15-00183-f018:**
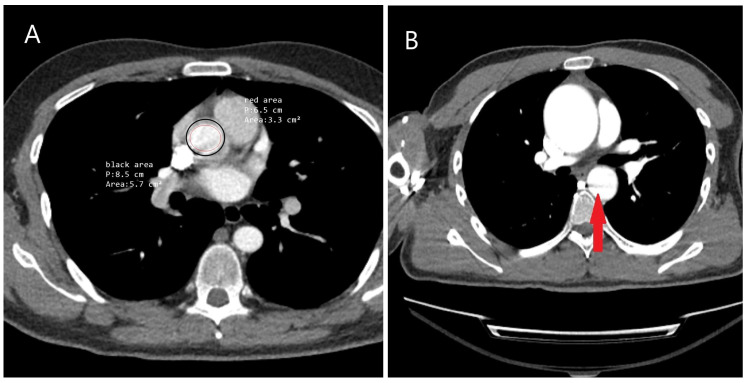
The significance of artifacts in CT imaging. (**A**) CT image of a 48-year-old male patient obtained to evaluate ascending aortic dilatation detected on echocardiography. The classic CT scan failed to clearly define the aortic diameter, with a discrepancy observed between the area within the red circle and the area within the black circle. Measurement using ECG-gated CT revealed an ascending aortic diameter of 48 mm. The difference in aortic diameters during systole and diastole caused this artifact. (**B**) CT image of a 30-year-old male patient with Marfan syndrome, showing a motion artifact in the descending aorta (red arrow), which is significant as it may mimic suspicion of dissection.

**Table 1 diagnostics-15-00183-t001:** Type and entry malperfusion classification used in aortic dissection.

Classification	Description
**Type (T)**	Type A, type B, non-A, non-B
**Entry Tear (E)**	**E0:** No entry tear **E1:** Entry tear in the ascending aorta (the area from the aortic valve to the proximal brachiocephalic trunk) **E2:** Entry tear in the aortic arch (the area from the brachiocephalic trunk to the left subclavian artery) **E3**: The aorta distal to the left subclavian artery
**Malperfusion (M)**	**M0:** No radiological or clinical evidence of malperfusion **M1:** Cardiac malperfusion (M1+: with ischemic changes, M1-: without ischemic changes) **M2:** Supra-aortic malperfusion (with or not cerebral or upper extremity) **M3:** Visceral or lower extremity malperfusion; dissection or false lumen origin of at least one visceral, renal, or iliac artery, or collapse of the aortic true lumen leading to functional closure of at least one branch of the visceral, renal, or iliac arteries

**Table 2 diagnostics-15-00183-t002:** Commonly used biomarkers in acute aortic syndromes.

Biomarker	Study (Year/Design)	Participant	Cutoff Value	Type (Diagnostic/Prognostic)	Remarks
Homocysteine (μmol/L)	Giusti et al. [46] (2003) Prospective	107 Marfan 189 healthy	13.5	Both	In hyperhomocysteinemic Marfan patients, the risk of aortic dissection was found to be approximately 4.5 times higher (OR 4.46, 95% CI 1.30–15.27, *p* = 0.017)
	Sbarouni et al. [47] (2013) Prospective	31 acute AD 30 AsAA 20 healthy	22	Diagnostic	The values in cases of acute dissection were 22 ± 14 compared to 17 ± 5 in chronic aortic aneurysm (*p* = 0.05)
	Golledge et al. [48] (2023) Prospective	471 with AbAA (30 to 54 mm)	15	Prognostic	In patients with small abdominal aortic aneurysms, the risk of major adverse cardiovascular events (MACE) is higher, with each 5.6 μM increase in homocysteine associated with a 29% increase in MACE risk (OR: 1.29, 95% CI: 1.13–1.48, *p* < 0.001)
IL-6 (pg/mL)	Wen et al. [49] (2012) Prospective	64 acute AD 42 chronic AD 98 hypertension 96 healthy	10.9	Diagnostic	3.3-fold increase in acute dissection compared to healthy individuals (*p* < 0.05)
	Wu et al. [50] (2020) Retrospective	141 type A AD	>108	Prognostic	a marker of early poor postoperative prognosis (area under the curve 0.901, 95% CI; 0.839–0.963), especially in combination with high D-dimer levels
	Chen et al. [51] (2022) Retrospective	331 type A AD	>259	Prognostic	Peak IL-6 > 259 pg/mL was identified as an independent risk factor for 30-day mortality; however, it was not predictive of renal injury
Creatine kinase-BB isozyme (IU/L)	Suzuki et al. [52] (1997) Prospective	10 acute AD 20 control	3.4	Diagnostic	It was measured at higher concentrations in aortic dissections (a 7.8-fold), and peak concentrations were observed 6-12 h after the onset of the disease
Smooth muscle myosin heavy chain (µg/L)	Suzuki et al. [53] (2020) Cross-Sectional	95 acute AD 48 AMI 131 healthy	2.5	Diagnostic	High in patients with acute AD presenting within 3 h, 25-fold increase compared to healthy individuals with 90.9% sensitivity. The test showed 98% specificity and 96% accuracy at a 2.5 mg/L threshold for aortic dissection diagnosis
Calponin (smooth muscle troponin-like protein; ng/mL)	Suzuki et al. [54] (2008) Prospective	59 acute AD 158 control	2,8 (acidic) 159 (basic)	Diagnostic	Optimal acidic calponin values showed 50% sensitivity and 87% specificity at 2.8 ng/mL, while basic calponin values showed 63% sensitivity and 73% specificity at 159 ng/mL (for 6 h). Acidic and basic calponins increased over two- and three-fold, respectively, within the first 6 h in AD
	Lian et al. [55] (2023) Retrospective	49 AAS 130 non-AAS	6.96 (acidic)	Diagnostic	In AAS, a two-fold increase compared to non-AAS was observed, with a cutoff value of 6.96 ng/mL (under the ROC curve: 88.9%)
Matrix metalloproteinases (ng/mL)	Koullias et al. [56] (2004) Histopathologic analysis	30 thoracic aneurysm 17 dissection 7 young cadavers		Diagnostic	The aortic walls in patients with dissection exhibit significantly higher levels of MMP2 and MMP9 expression compared to those with non-dissecting aneurysms
	Wen et al. [49] (2012) Prospective	64 acute AD 42 chronic AD 98 hypertension 96 healthy	37.7 55.7 107.2	Both	MMP9 concentrations are higher in patients with chronic AD (55.7) than acute AD (37.7) and increase immediately after surgical treatment or stenting. Among treated patients who died, the level was notably high (107.2)
	Giachino et al. [57] (2013) Prospective	52 acute AD 74 non-AD	3.6 (MMP8) 20 (MMP9)	Diagnostic	In acute AD, MMP8 levels were 2.75 times higher, and MMP9 levels were 2 times higher. At a cutoff value of 11.0 ng/mL, the negative predictive value of MMP8 reaches 100% when used in combination with D-dimer
	Proietta et al. [58] (2014) Prospective	23 AD 21 CAS 21 CVRF 10 healthy	20.4	Diagnostic	MMP12 shows a sixfold increase in AD patients compared to healthy individuals and is a potential biomarker for AD, especially in those without genetic predisposition
	Zhang et al. [59] (2014) Case–control	25 DTAA 17 organ-donor		Diagnostic	Dissection tissue exhibits increased levels of total MMP1 and MMP9, decreased MMP2, and immunostaining revealing higher expression of MMP-1, -3, -9, -12, and -13 in the media of the false lumen’s outer aortic wall compared to control tissue
	Vianello et al. [60] (2016) Prospective	22 type A AD 11 type B AD 30 healthy	1.5–2	Diagnostic	In acute AD, MMP1 levels are elevated, particularly in type A and within the first 24 h, whereas MMP2 and MMP9 do not show significant increases compared to controls
	Zhang et al. [61] (2017) Retrospective	72 AbAA 72 HT 72 healthy	2.8	Diagnostic	In hypertensive patients with AbAA patients, the blood level of MMP7 is 5.5 times higher than in healthy individuals and is highly expressed in diseased aortic tissue
	Li et al. [62] (2018) Case–control	88 AD 88 healthy	379.4	Diagnostic	AUC for MMP9: 0.810 (sensitivity: 68.2%, specificity: 84.1%); elevated in AD, correlated with CRP, not with D-dimer
	Jia et al. [63] (2023) Prospective	155 acute AD	16.6	Both	High MMP9 levels are both an independent risk factor for mortality and a significant positive correlation with aortic diameter and false lumen area
	Irqsusi et al. [64] (2024) Histopathologic analysis	52 AD 52 AsAA 7 CAD		Diagnostic	Higher expression of MMP1 and MMP9 is observed in the adventitia, and elevated MMP9 levels in the media, particularly in patients with AD
Soluble suppression of tumorigenesis-2 (sST2); (IL)–1 receptor family member (ng/mL)	Wang et al. [65] (2018) Hybrid cohorts	1027 retrospective 333 prospective	34.6	Diagnostic	High in acute dissection compared to AMI and PE; sensitivity of 99.1%, specificity of 84.9%. Using sST2 at around 35 ng/mL can exclude aortic dissection with a negative likelihood ratio of <0.1 and a negative predictive value of >90%
	Morello et al. [66] (2020) Prospective	88 AAS 209 non-AAS	Different levels	Diagnostic	It is elevated in AAS patients, but in ROC analysis, the AUC of sST2 was 0.63 (D-dimer: 0.82, *p* < 0.001), indicating modest accuracy, with sensitivity ranging from 35.2 to 95.5% and specificity from 10.8 to 85.1% across different cutoffs
	Zhu et al. [67] (2024) Retrospective	90 type B AD 92 IMH 90 non-AAS	27.54	Diagnostic	sST2 levels increase with the onset of type B AD (sensitivity of 80.92% and specificity of 75.00%), but its combination with D-dimer demonstrates strong diagnostic performance in intramural aortic hematoma (sensitivity, 69.20%; specificity, 80.00%)

AAS, acute aortic syndrome; AbAA, abdominal aortic aneurysm; AD, aortic dissection; AMI, acute myocardial infarction; AsAA, ascending aortic aneurysm; AUC, area under the curve; CAD, coronary artery disease; CAS, carotid artery stenosis; CI, confidence interval; CVRF, cardiovascular risk factors; DTAA, descending thoracic aortic aneurysm; HT, hypertension; IMH, intramural hematoma; PE, pulmonary embolism; ROC, receiver operating characteristic; OR, odds ratio.

**Table 3 diagnostics-15-00183-t003:** Characteristics of imaging techniques used in the evaluation of aortic diseases [4,5,6,7,24,36].

Imaging Modality	Sensitivity (%)	Specificity (%)	Advantages	Disadvantages
Computed Tomography (CT)	95–100	98–99	Rapid, widely available, high resolution. Provides detailed anatomical information for dissection and rupture.	Radiation exposure, contrast nephropathy, or anaphylaxis risk. Motion artifacts can affect image quality.
Magnetic Resonance Imaging (MRI)	97–100	94–100	No radiation exposure, excellent soft tissue contrast. Ideal for soft tissue abnormalities such as intramural hematomas.	Limited availability, time-consuming, and contraindicated in unstable patients. Expensive infrastructure requirements.
Transthoracic Echocardiography (TTE)	60–90 (for ascending aorta) 30–60 (for descending aorta)	80–96 60–80	Non-invasive, widely available. Useful for initial evaluation and bedside use.	Limited accuracy for distal aorta, operator-dependent. Poor imaging in obese or chronic obstructive pulmonary disease patients.
Transesophageal Echocardiography (TEE)	86–100	90–100	High resolution for proximal aorta, accessible in critical settings. Effective for real-time blood flow assessment.	Semi-invasive, requires sedation or anesthesia. Cannot be used in patients with esophageal pathology.
Chest X-Ray (CXR)	30–50	60–70	Low-cost, quick, initial screening tool. Often used to rule out obvious abnormalities.	Low sensitivity and specificity, non-diagnostic in many cases. Limited ability to assess complex aortic pathology.
Positron Emission Tomography (PET)	85–95	80–90	Detects inflammatory and infectious processes, functional imaging. Useful for vasculitis and other systemic diseases and can detect inflammation in aortic syndromes.	Expensive, limited availability, requires radioactive tracers. Long preparation and imaging times.
Invasive Aortography	85–95	>95	Gold standard for vascular imaging. Provides detailed lumen and vascular anatomy with high resolution.	Invasive procedure requiring arterial access. Associated with risks like bleeding, infection, and contrast nephropathy. Not provide any information on aortic wall thickness.

**Table 4 diagnostics-15-00183-t004:** Impact of metabolites on aortic pathophysiology and potential treatment.

Metabolite	Effect	Potential Treatment	References
Hyperhomocysteinemia	- Synthetic VSMC formation - Decrease in nitric oxide production - Increase in elastin-degrading enzymes - Increase in MMP-2 and MMP-9 - Release of inflammatory cytokines - Increase in ROS production - Thromboxane production and platelet activation - Increase in endothelin (H-type hypertension) - Increase in C-reactive protein	Vitamin B6 Vitamin B12 Folic acid Antioxidants (vitamin E and C/polyphenols) Taurine Probucol	Balint et al. [114] Huang et al. [115] Van Hove et al. [116] Chen et al. [117] Li et al. [118]
Kynurenine–tryptophan pathway	- Increase in nicotinamide adenine dinucleotide phosphate oxidase - Increase in ROS production - Endothelial damage - Apoptosis	Kynureninase Melatonin	Wang et al. [119] Xia et al. [120]
Impaired lipid metabolism	- Foam cell formation - Synthetic VSMC transformation - Caspase 1 activation and cell death with oxidized LDL - Increase in chemokines - Increase in ROS - Atherogenic	Statins Long-chain omega-3 polyunsaturated fatty acid Nitro-oleic acid	Kattoor et al. [121] Jovin et al. [122] Meital et al. [123] Nettersheim et al. [124]
Glycolytic activity	- Warburg effect - Lactate and lactate dehydrogenase-A activation - Increase in synthetic VSMC - Increase in endothelial permeability - Increased amount of hypoxic tissue	Glycolysis inhibitor 2-deoxyglucose (2-DG) Small interfering RNA (siRNA)	Tsuruda et al. [125] Kim et al. [126]
Amyloid	- Pro-inflammatory through angiotensin II - Aging accelerator - Increase in MMP-2/-9 - Medial degeneration	A treatment protocol is determined based on the location of involvement (e.g., colchicine, melphalan, steroids, etc.).	He et al. [127] Peng et al. [128] Merlini et al. [129]
Succinate	- Increase in ROS via succinate dehydrogenase - Stimulation of cytokine release - Formation of hypoxic regions - Exacerbation of ischemia–reperfusion injury - Could be used as a biomarker for pre-diagnosis	Dimethyl malonate	Cui et al. [130] Chouchani et al. [131]

LDL, low-density lipoprotein; MMP, matrix metalloproteinase; ROS, reactive oxygen species; VSMC, vascular smooth muscle cell.

## Data Availability

The imaging materials used in this review are stored in the authors’ archives, and no materials have been obtained from external sources or other publications. Imaging materials are available upon request from the corresponding author.

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
