# Peer review of "A Narrative Review of Biomarkers and Imaging in the Diagnosis of Acute Aortic Syndrome"

_diagnostics, 2025, doi:10.3390/diagnostics15020183_

Round 1

Reviewer 1 Report

Comments and Suggestions for Authors

I read with interest the Review article by Ümit Arslan and Ä°zatullah Jalalzai titled: A Review of Biomarkers and Imaging in the Diagnosis of Acute Aortic Syndrome.

I should congratulate the authors for this comprehensive but still extensive overview of these diseases.

The manuscript is well-written and accompanied by Tables that help convey important messages.

I do not have any major comments and although the literature on this condition is quite extensive I would support the publication of this article as this is a nice addition to the already existing ones.

Author Response

Comment:

I read with interest the Review article by Ümit Arslan and Ä°zatullah Jalalzai titled: A Review of Biomarkers and Imaging in the Diagnosis of Acute Aortic Syndrome.

I should congratulate the authors for this comprehensive but still extensive overview of these diseases.

The manuscript is well-written and accompanied by Tables that help convey important messages.

I do not have any major comments and although the literature on this condition is quite extensive I would support the publication of this article as this is a nice addition to the already existing ones.

answer:

Dear Reviewer,

We sincerely thank you for your thoughtful and positive comments on our manuscript titled "A Review of Biomarkers and Imaging in the Diagnosis of Acute Aortic Syndrome." Your recognition of our efforts to provide a comprehensive yet concise overview of these diseases is truly encouraging.

We are especially pleased to hear that the tables effectively conveyed the key messages, as we aimed to make the information accessible and organized for the readers.

Your support for the publication of our work is greatly appreciated, and your feedback motivates us to continue contributing to the literature in this field.

Thank you once again for your time and valuable insights.

Best regards,

Reviewer 2 Report

Comments and Suggestions for Authors

I read with interest the review titled (A Review of Biomarkers and Imaging in the Diagnosis of Acute Aortic Syndrome). The manuscript tackle an important research area and needed clinically.

The title is expressive, while it is necessary to determine the type of the review.

The abstract adequately summarizes the review, however, it is required to mention the focus of the review, what biomarkers and imaging will be reviewed.

What are the sources of the images used?

Provide a brief description of the search methodology.

 In table 2: provide a column for the type of the marker (diagnostic vs. prognostic)

Expand of the types of MMP used and include more recent studies.

A section was dedicated to D-dimer only, please discuss other biomarkers in more details in the text.

Table 3 needs references.

The article could benefit from a dedicated section on future research directions or unanswered questions in the field of AAS and its diagnosis.

The manuscript could benefit from improved flow between sections. Some transitions feel abrupt, and the connection between different topics could be made clearer. Genetics appeared suddenly, please indicate in the review and objectives that you will discuss genetics.

AI was mentioned in the abstract, however, it was discussed lightly in the text. Either remove it from abstract of provide a section that deeply discuss the role of AI in acute aortic syndrome.

Discuss the AAS in the settings of autoimmune aortitis.

Remove citations from the conclusions. The manuscript lacks a strong conclusion that summarizes the key points and highlights the future directions of research in AAS diagnosis.

Provide recommendations for future research directions. 

Author Response

Dear Reviewer 2

We sincerely thank you for taking the time to thoroughly review our manuscript. Your insightful comments and constructive suggestions have provided us with valuable guidance to enhance the clarity, structure, and overall quality of our work. We hope that the revisions meet your expectations and address your concerns effectively. Best Regards;

Below, we have addressed each of your comments in detail: 

  1. Comment The title is expressive, while it is necessary to determine the type of the review.

Answer: Thank you for your suggestion. We have clarified the type of the review in the title by updating it to: "A Narrative Review of Biomarkers and Imaging in the Diagnosis of Acute Aortic Syndrome."

  1. Comment: The abstract adequately summarizes the review, however, it is required to mention the focus of the review, what biomarkers and imaging will be reviewed.

Answer: Thank you for your valuable suggestion. The abstract has been revised to explicitly mention the focus of the review, including key biomarkers as well as imaging modalities

  1. Comment What are the sources of the images used?

Answer: Thank you for your question. As stated in the "Data Availability Statement" section of the manuscript, all images are sourced from the authors' archives. No materials have been obtained from external sources or other publications.

  1. Comment. Provide a brief description of the search methodology.

Answer: Thank you for your valuable comment. We have added a "Methodology" section to the manuscript to describe the literature search process.

  1. Comment In table 2: provide a column for the type of the marker (diagnostic vs. prognostic)

Answer: Thank you for your valuable suggestion. We have updated Table 2 to include a column specifying the type of each marker (diagnostic or prognostic). The classification was determined based on how the biomarkers were used in the studies cited in the table.

  1. Comment Expand of the types of MMP used and include more recent studies.

Answer: Thank you for your valuable suggestion. We reviewed recent studies on MMPs and highlighted those involving MMP-2 and MMP-9, as well as studies focusing on other MMPs, which are summarized in Table 2

  1. Comment A section was dedicated to D-dimer only, please discuss other biomarkers in more details in the text.

Answer: Thank you for your valuable suggestion. In response, we have expanded the manuscript to include additional biomarkers. Biomarkers such as homocysteine and BNP, which are widely measurable in clinical practice, have been discussed under separate headings. Other less commonly used or experimental biomarkers are addressed in a distinct section to provide clarity and structure. We believe these additions fulfill your request and enhance the manuscript's comprehensiveness.

  1. Comment Table 3 needs references.

Answer: Thank you for your valuable comment. We have added the references from which the data were derived.

  1. Comment The article could benefit from a dedicated section on future research directions or unanswered questions in the field of AAS and its diagnosis.

Aswer: Thank you for your insightful suggestion. In response, we have included a dedicated section on future directions,

  1. Comment The manuscript could benefit from improved flow between sections. Some transitions feel abrupt, and the connection between different topics could be made clearer. Genetics appeared suddenly, please indicate in the review and objectives that you will discuss genetics.

Answer. Thank you for bringing this important point to our attention with your valuable feedback. In response, we have revised the manuscript to improve the flow by relocating the genetics-related and aortitis sections under the "Pathological Definitions" heading.

  1. Comment: AI was mentioned in the abstract, however, it was discussed lightly in the text. Either remove it from abstract of provide a section that deeply discuss the role of AI in acute aortic syndrome.

Answer: Thank you for your insightful observation. We have removed the mention of AI from the abstract as per your suggestion. However, we have retained a brief mention of AI in the text to inform readers about its emerging role and the ongoing research in this area. While a detailed discussion of AI is beyond the scope of this review, we believe that highlighting its potential in acute aortic syndrome may inspire curiosity and guide readers toward exploring the relevant studies

  1. Comment Discuss the AAS in the settings of autoimmune aortitis.

Answer: Thank you for your valuable suggestion. As requested, we have briefly addressed autoimmune aortitis in the context of AAS, supported by case examples and references to relevant studies.

  1. Comment Remove citations from the conclusions. The manuscript lacks a strong conclusion that summarizes the key points and highlights the future directions of research in AAS diagnosis.

Answer: Thank you for your valuable feedback. In response, we have removed citations from the conclusion and revised it to provide a concise summary of the key points while emphasizing future research directions. We hope these revisions sufficiently address your comment.

  1. Comment Provide recommendations for future research directions. 

Answer: Thank you for your insightful suggestion. In response, we have included a dedicated section on future research directions, addressing unanswered questions in the field

Round 2

Reviewer 2 Report

Comments and Suggestions for Authors

Thank you for responding to my comments